# Ensuring scientific reproducibility in bio-macromolecular modeling via extensive, automated benchmarks

Julia Koehler Leman [1,2,34✉], Sergey Lyskov [3,34], Steven M. Lewis[4,34], Jared Adolf-Bryfogle [5,6], Rebecca F. Alford[3], Kyle Barlow [7], Ziv Ben-Aharon [8], Daniel Farrell [9,10], Jason Fell [11,12,13], William A. Hansen[14,15], Ameya Harmalkar [3], Jeliazko Jeliazkov [16], Georg Kuenze[17,18,19], Justyna D. Krys [20], Ajasja Ljubetič [9,10], Amanda L. Loshbaugh[21,22], Jack Maguire[23], Rocco Moretti[17,18], Vikram Khipple Mulligan[1], Morgan L. Nance [16], Phuong T. Nguyen[24], Shane Ó Conchúir[21], Shourya S. Roy Burman [3], Rituparna Samanta[3], Shannon T. Smith [18,25], Frank Teets[26], Johanna K. S. Tiemann [27], Andrew Watkins[28], Hope Woods [18,25], Brahm J. Yachnin [14,15], Christopher D. Bahl[29,30,31], Chris Bailey-Kellogg[32], David Baker [9,10], Rhiju Das [28], Frank DiMaio[9,10], Sagar D. Khare[14,15], Tanja Kortemme[21,22], Jason W. Labonte[3], Kresten Lindorff-Larsen [27], Jens Meiler [17,18,19], William Schief [5,6], Ora Schueler-Furman [8], Justin B. Siegel[11,12,13], Amelie Stein [27], Vladimir Yarov-Yarovoy [24], Brian Kuhlman [26], Andrew Leaver-Fay [26], Dominik Gront[20], Jeffrey J. Gray [3✉] & Richard Bonneau [1,2,33✉]

Each year vast international resources are wasted on irreproducible research. The scientific community has been slow to adopt standard software engineering practices, despite the increases in high-dimensional data, complexities of workflows, and computational environments. Here we show how scientific software applications can be created in a reproducible manner when simple design goals for reproducibility are met. We describe the implementation of a test server framework and 40 scientific benchmarks, covering numerous applications in Rosetta bio-macromolecular modeling. High performance computing cluster integration allows these benchmarks to run continuously and automatically. Detailed protocol captures are useful for developers and users of Rosetta and other macromolecular modeling tools. The framework and design concepts presented here are valuable for developers and users of any type of scientific software and for the scientific community to create reproducible methods. Specific examples highlight the utility of this framework, and the comprehensive documentation illustrates the ease of adding new tests in a matter of hours.

A full list of author affiliations appears at the end of the paper.

Reproducibility in science is a systemic problem. In a survey published by *Nature* in 2016, 90% of scientists responded that there is a reproducibility crisis[1]. Over 70% of the over 1500 researchers surveyed were unable to reproduce another scientist's experiments and over half were unable to reproduce their own experiments. Another analysis published by *PLOS One* in 2015 concluded that, in the US alone, about half of preclinical research was irreproducible, amounting to a total of about $28 billion being wasted per year[2]!

Reproducibility in biochemistry lab experiments remains challenging to address, as it depends on the quality and purity of reagents, unstable environmental conditions, and the accuracy and skill with which the experiments are performed. Even small changes in input and method ultimately lead to an altered output. In contrast, computational methods should be inherently scientifically reproducible since computer chips perform computations in the same way, removing some variations that are difficult to control. However, in addition to poorly controlled computing environment variables, computational methods become increasingly complex pipelines of data handling and processing. This effect is further compounded by the explosion of input data through "big data" efforts and exacerbated by a lack of stable, maintained, tested, and well-documented software, creating a huge gap between the theoretical limit for scientific reproducibility and the current reality[3].

These circumstances are often caused by a lack of best practices in software engineering or computer science[4,5], errors in laboratory management during project or personnel transitions, and a lack of academic incentives for software stability, maintenance, and longevity[6]. Shifts in accuracy can occur when re-writing functionality or when several authors work on different parts of the codebase simultaneously. An increase in complexity of scientific workflows with many and overlapping options and variables can prevent scientific reproducibility, as can code implementations that lack or even prevent suitable testing[4]. The absence of testing and maintenance causes software erosion (also known as *bit rot*), leading to a loss of users and often the termination of a software project. Further, barriers are created through intellectual property agreements, competition, and refusal to share inputs, methods, and detailed protocols.

As an example, in 2011 the Open Science Collaboration in Psychology tried to replicate the results of 100 studies as part of the Reproducibility Project[7]. The collaboration consisting of 270 scientists could only reproduce 39% of study outcomes. Since then, some funding agencies and publishers have implemented data management plans or standards to improve reproducibility[8–11], for instance, the FAIR data management principles[12]. Guidelines to enhance reproducibility[13,14] are certainly applicable, are outlined in Table 3, and are discussed in detail in an excellent editorial[15] describing the *Ten Year Reproducibility Challenge*[16] that is published in its own reproducibility journal ReScience C[17]. Other efforts focus directly on improving the methods with which the researchers process their data—for instance, the Galaxy platform fosters accessibility, transparency, reproducibility, and collaboration in biomedical data analysis and sharing[13].

Reproducibility is also impacted by *how* methods are developed. Comparing a newly developed method to established ones, or an improved method to a previous version is important to assess its accuracy and performance, monitor changes and improvements over time and evaluate the cost/benefit ratio for software products to commercial entities. However, biases in publishing positive results or improvements to known methods, in conjunction with errors in methodology or statistical analyses[18], lead to an acute need to test methods via third parties. Often, methods are developed and tested on a specific benchmark set created for that purpose and will perform better on that dataset than methods not trained on that dataset. A rigorous comparison and assessment require the benchmark to be independently created from the method, which unfortunately is rarely the case. Compounding issues are lack of diversity in the benchmark set (towards easier prediction targets) and reported improvements smaller than the statistical variation of the predicted results. Guidelines on how to create a high-quality benchmark[19,20] are outlined in Table 3 below.

Scientific reproducibility further requires a stable, maintainable, and well-tested codebase. Software testing is typically achieved on multiple levels[4,21]. Unit tests check for scientific correctness of small, individual code blocks, integration tests check an entire application by integrating various code blocks, and profile and performance tests ensure consistency in runtime and program simplicity. Scientific tests or benchmarks safeguard the scientific validity and accuracies of the predictions. They are typically only carried out during or after the development of a new method (static benchmarking), as they require domain expertise and rely on vast computational resources to test an application on a larger dataset. However, the accuracy and performance of a method depend on the test set, the details of the protocol (i.e., specific command lines, options, and variables), and the software version. To overcome the static benchmarking approach, blind prediction challenges such as the Critical Assessments in protein Structure Prediction[22], PRediction of protein Interactions[23], Functional Annotation[24], Genome Interpretation[25], RNA Puzzles[26], and Continuous Automated Model EvaluatiOn[15,27] hold double-blind competitions at regular intervals. While these efforts are valuable to drive progress in method development in the scientific community, participation often requires months of commitment and does not address the reproducibility of established methods over time.

The Rosetta macromolecular modeling suite[28,29] has been developed for over 20 years by a global community with now hundreds of developers at over 70 institutions[4,30]. This history and growth required us to adopt many best practices in software engineering[4,29], including the implementation of a battery of tests. A detailed description of our community, including standards and practices, has previously been provided[4]. Scientific tests are important to maintain prediction accuracies for our own community and our users (including commercial users whose licensing fees, in our case, support much of Rosetta's infrastructure and maintenance). We further want to directly compare different protocols and implementations and monitor the effect of score function changes on the prediction results. For many years, Rosetta applications[31,32] and score functions[33–36] have been tested independently using the static benchmarking approach[20,37], often with complete protocol captures[38,39]. The disadvantage of static benchmarking is that the results become outdated due to the lack of automation. Reproducibility becomes impossible due to a lack of preservation of inputs, options, environment variables, and data analyses over time.

This background highlights the challenges in rigorously and continuously testing how codebase changes affect the scientific validity of a prediction method while maintaining or improving scientific reproducibility. Running scientific benchmarks continuously (1) suffers from a lack of incentive to set up as the maintenance character of these tests collides with academic goals; (2) requires both scientific and programming/technical expertise to implement, interpret and maintain; (3) is difficult to interpret with pass/fail criteria; and (4) requires a continuous investment of considerable computational resources. Here, we address these challenges by introducing a general framework for continuously running scientific benchmarks for a large and increasing number of protocols in the Rosetta macromolecular modeling suite.

We present the general setup of this framework, demonstrate how we solve each of the above challenges, and present the results of the individual benchmarks in the Supplementary Information of this paper, complete with detailed protocol captures. The results can be used as a baseline by anyone developing macromolecular modeling methods, and the code of this framework is sufficiently general to be integrated into other types of software. The design principles presented here can be used by anyone developing scientific software, independent of the size of the method. We highly encourage small software development groups to follow these guidelines, even though their technical and personnel setup might differ. Supplementary Note (1) describes several options that small groups have available to test their software with limited resources.

## Results

Successful software development can be achieved by following a number of guidelines which are have previously been described in detail in ref. [4]. Software testing is an essential part of this strategy which ties into scientific reproducibility. Over the past 15 years, the Rosetta community has created its own custom-built test server framework connected to a dedicated high-performance computing (HPC) cluster—its setup is shown in Fig. 1A and described in the Supplementary Information. The scientific testing setup is integrated into this framework.

**Insights from the previous round of scientific tests led to specific goals**. The Rosetta community learned valuable lessons from the long-term maintenance (or lack thereof) of several scientific benchmark tests set up over 10 years ago (see

Supplementary Note 2). Their deterioration and development life cycle motivated specific goals that we think lead to more durable scientific benchmarks (Fig. 1B): (1) simplicity of the framework to encourage maintenance and support; (2) Generalization to support all user interfaces to the Rosetta codebase (command line, RosettaScripts[40], PyRosetta[41,42]); (3) automation to continuously run the tests on an HPC cluster with little manual intervention; (4) documentation on how to add tests and scientific details of each test to allow maintenance by anyone with a general science or Rosetta background; (5) distribution of the tests to both the Rosetta community and their users, and publicizing their existence to encourage the addition of new tests and maintenance by the community; and (6) maintenance of the tests, facilitated by each of the previous points.

**Goal 1—simplicity: simple setup facilitates broad adoption and support from our community**. To encourage our community to contribute as many tests as possible, the testing framework needs to be simple and support fast and easy addition of tests. We decided on a Python framework that integrates well with our pre-existing testing HPC cluster (Supplementary Note 3). We further require these tests to be able to run on local machines (with different operating systems) as well as various HPC clusters with minimal adjustments. Debugging the scripts should be as simple as possible. With these requirements in mind, we decided on a setup as shown in Fig. 1C. We provide a template directory with all necessary files (described in detail in *Methods*). Simple modifications like naming scripts in the order in which they run—e.g., *0.compile.py* to *9.finalize.py*—greatly facilitate debugging or extension by new users.

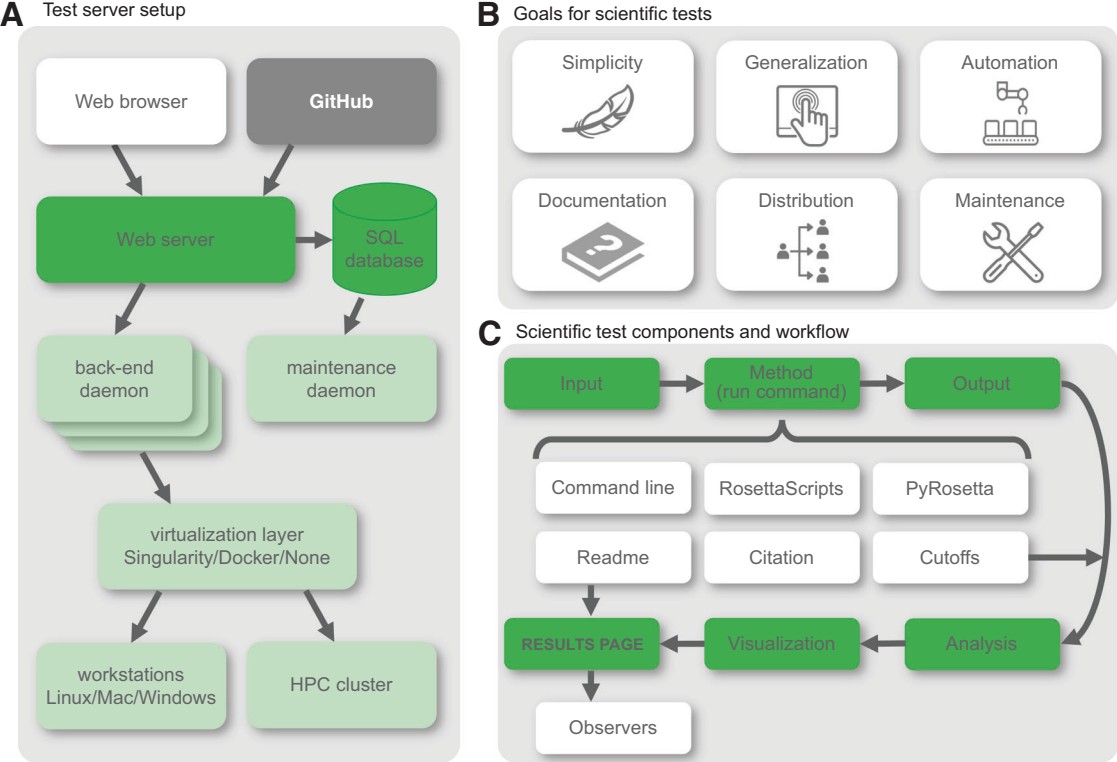

**Fig. 1 Goals and setups for the scientific tests. A** Test server setup with the web browser as the user interface, the frontend in bright green, and the backend in light green. The code is stored in GitHub, shown in dark gray. **B** Specific goals for our scientific tests, driven by flaws in a previous iteration of these tests. Each point is described in detail in the text. **C** Basic infrastructure of the scientific test framework, motivated by simplicity. Each box represents a file, folder, or script that is either provided in the template folder or generated throughout the protocol run. The basic workflow is highlighted in green with components that facilitate documentation and maintenance shown in white. [Icons in Fig. 1B were created by Ana Teixeira, Aman, Ben Davis, Gregor Cresnar, Anna Sophie, and Joel Avery from Noun Project.] SQL structured query language, HPC cluster high-performance computing cluster.

**Goal 2—generalization: new tests support interfaces of the command line, PyRosetta, or RosettaScripts.** Rosetta supports several interfaces to facilitate quick protocol development while lowering the necessary expertise required by new developers to join our community[4]. Many mainstream protocols have been developed as standalone applications to be run via the command line, while customized protocols have been developed in RosettaScripts[40] and PyRosetta[41,42]. For our test server framework, we sought a general code design that allows input from all three interfaces while supporting different types of outputs, quality measures, and analyses, sometimes even written in different scripting languages.

**Goal 3—automation: tests require substantial compute power and are run on a dedicated test server.** Running scientific benchmarks requires extensive CPU time; hence we chose to integrate them with our own custom-built test server framework connected to a dedicated HPC cluster (Fig. 1A and Supplementary Information). This test server framework consists of two main components: the backend holds low-level primitive code for compilation on different operating systems and HPC environments, cluster submission scripts, and web server integration code. The front end contains the test directories that are implemented by the test author. Our test server is accessible through a convenient web interface (Fig. 2A; available at https://benchmark.graylab.jhu.edu/). This framework has had a hugely

positive impact on the growth and maintenance of both the Rosetta software and our community, due to its accessibility, GitHub integration, ease of use, and automation. In small software communities that lack the ability or resources to set up a dedicated test server, integration testing via external services like Github Actions[43], Drone CI[44], Travis CI[45], or Jenkins[46] is an excellent alternative. More details can be found in the Supplementary Information.

The RosettaCommons supports our benchmarking effort through expansion of our centralized test server cluster hardware and labor with an annual budget (see Supplementary Information and our previous publication[4]). Because the scientific tests are integrated into our test server framework, authors of the tests can focus on the scientific protocols (starting from a template directory set up as in Fig. 1C) instead of debugging errors in compilation, cluster submission, and computational environment. This pattern also makes these tests system-independent (the author writes the setup for a local machine and runs it on this server), i.e., portable between operating systems and computational environments. We currently limit the runtime per scientific test to typically 1000–2000 CPU hours.

Due to the required computational resources, we are unable to test every code revision in the main development branch of Rosetta; instead, we dedicate computational nodes to the scientific tests and run tests such that the nodes are continuously occupied. We found that scheduling the earliest-run test on an individual rolling basis, as compute nodes become available, is most efficient

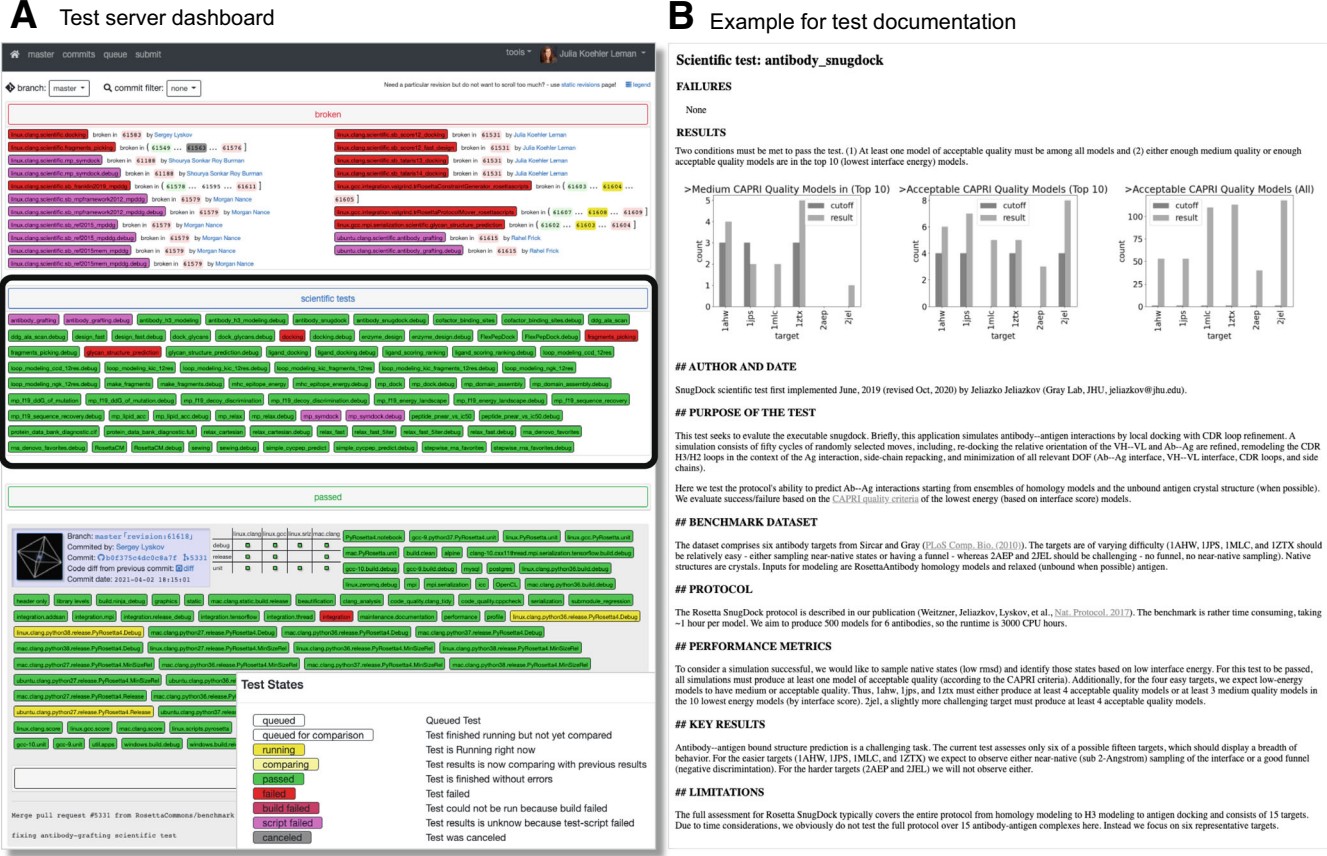

**Fig. 2 Webpages for the main dashboard and documentation of the tests. A** Dashboard of our benchmark server testing infrastructure. Each test is colored according to its test results: red denotes breakage, magenta denotes script failure, green denotes passing of a test, yellow denotes the test is currently running, and white denotes the test has yet to be run. All broken tests are shown prominently at the top of the page. All scientific tests are shown in the blue tab below (also encircled in bold black). Tests of the latest revision merged into the main branch are shown below with information about the committer, the pull request ID, a link to the code difference, and the commit message. **B** The results page shows the results of the run, the documentation, and the description of whether the test passes or fails. Results pages are automatically generated at the end of the run for each test.

in balancing the server load while keeping nodes available for tests in feature branches. Upon discovery of a test failure and to find the specific revision (and therefore the code change) that caused the failure, our *bisect* tool schedules intermediate revisions on a low-priority basis. All test results are stored in the database and are accessible through a web interface (Fig. 2).

**Goal 4—documentation: anyone can quickly and easily add new tests**. Creating well-designed scientific benchmarks requires expertise in defining the scientific objective, establishing a protocol, and creating a high-quality test dataset. The last step of incorporating the test into our framework should be as simple as possible (as per our *simplicity* requirement). Once the dataset, interface (command line, RosettaScripts, or PyRosetta), specific command line, and quality measures have been chosen, the author can simply follow the individual steps outlined on the documentation page[47] to contribute the test; the template guides the setup (Supplementary Note 4). We found that the setup is simple enough that untrained individuals can contribute a test in a few hours based on documentation alone—hence we achieved our goal of simplicity and detail in our documentation.

One of the reasons for the deterioration of earlier scientific tests was lack of maintenance due to insufficient documentation. Our goal is to drive the creation of extensive documentation for each test such that anybody with an average scientific knowledge of biophysics and introductory knowledge of programming in Rosetta can understand and maintain the tests. To ensure comprehensive documentation and consistency between tests, we provide a readme template with specific sections and questions that need to be answered for each test (see Supplementary Information). The template discourages writing short, insufficient, free-form documentation, and instead encourages the addition of important details and significantly lowers the barrier for writing extensive documentation. The questionnaire-style readme template (see Supplementary Information) saves time to locate necessary details to repair broken tests. The extent and quality of documentation is independently approved by a pull-request reviewer before the test is merged into the main repository. The benchmarking framework is configured such that documentation needs to be written once and is then directly embedded into the results page. Thus, the documentation is accessible both in the code and on the web interface while eliminating text duplication that could lead to discrepancies and confusion.

**Goal 5—distribution: additions and usage of tests by our community requires broad distribution**. Earlier scientific tests also deteriorated due to poor communication as to the existence of these tests, which resulted in a small pool of maintainers. Because our new scientific tests are integrated into our test server framework which most of our community uses and monitors, developers are immediately aware of the tests that exist and their pass/fail status. In conjunction with regular announcements to our community, this visibility should significantly broaden the number of people able and willing to sustain the scientific tests for a long time. If we nevertheless find that our new tests deteriorate, we will host a hackathon (eXtreme Rosetta Workshop[4]) to supplement or repair these tests in a concentrated effort.

**Goal 6—maintenance: test failures are handled by a defined procedure**. The often overlooked, *real* work in software development is not necessarily the development of the software itself, but its maintenance. We have a system in place outlining how test failures are handled and by whom. Each test has at least one dedicated maintainer (aka 'observer', usually the test author) who

is notified of the test breakage via email and whose responsibility it is to repair the test. Test failures can be three-fold: technical failures, stochastic failures, or scientific failures. Technical failures (such as compiler errors, script failures due to new versions of programs, etc.) typically require small adjustments and fall under the responsibility of the test author and our dedicated test engineer.

Stochastic failures are an uncommon feature in software testing and are a rare but possible occurrence in this framework. Rosetta often uses Metropolis Monte Carlo algorithms and thus has an element of randomness present in most protocols. Setting specific seeds is done for integration tests in Rosetta (which are technically regression tests, discussed in the Supplementary Information of a previous publication[4]), which are not discussed here in detail. We refrain from setting random seeds in our scientific tests because the goal is to check whether the overall statistical and scientific interpretations hold after running the same protocol twice, irrespective of the initial seed. Further, a change in the vast Rosetta codebase that adds or removes a random number generator call is expected to cause trajectory changes even with set random seeds. The scientific tests are scaled so that individual trajectories are treated statistically and the lack of response to both seed changes and minor code changes is a feature and goal of the test. Moreover, due to the reasons above, rare stochastic failures are not a concern in our case and point to a sensibly chosen cutoff value (Supplementary Note 5). Scientific tests are interpreted in a Boolean pass/fail fashion but generally have an underlying statistical interpretation and are sampling from a distribution against a chosen target value. The statistical interpretation often varies from test to test and depends on the output of the protocol, the types of quality metrics, and sample sizes; therefore, we cannot provide specific suggestions as to which statistical measures should be used in general. Details about which statistics are used in which protocol are provided in the Supplementary Information and the linked tests. The randomness of Monte Carlo will occasionally cause a stochastic test failure because those runs happen to produce poor predictions by the tested metric. This is handled by simply rerunning the test: rare "stochastic" failures are either not stochastic—i.e., the test is signaling breakage—or are a symptom that the structure or pass/fail criteria of the test are not working as intended.

A scientific failure requires more in-depth troubleshooting and falls under the responsibility of the maintainer. If the maintainer does not fix the test, we have a rank-order of responsibilities to enforce the test repair. The principal investigator of the test designates someone in their lab. If the necessary expertise does not exist in the lab at the time (usually because people have moved on in their career), repairing the test becomes the responsibility of the person who broke it. If this developer lacks the expertise, the repair becomes community responsibility, which typically falls onto one of our senior developers.

**Most major Rosetta protocols are now implemented as scientific benchmarks**. Using the framework described above, our community implemented 40 scientific benchmarks spanning a broad range of applications including antibody modeling, docking, loop modeling, incorporation of NMR data, ligand docking, protein design, flexible peptide docking, membrane protein modeling, etc. (Table 1 and Supplementary Note 6). Each benchmark is unique in its selection of targets in the benchmark set, the specific protocol that is run, the quality metrics that are evaluated, and the analysis to define the pass/fail criterion. The details for all the benchmarks are provided in the comprehensive supplement to this paper. We further publish the benchmarks

**Table 1 Scientific tests for bio-macromolecular modeling, continuously running on our testing server framework.**

| Test suite | Tests | Refs. | Test author | Quality measures | Targets | nstruct | Runtime in CPUh |
|---|---|---|---|---|---|---|---|
| Antibodies | antibody_grafting | 55 | Jeliazko Jeliazkov | Fraction residues within rmsd to native | 48 | 1 | 3 |
| | antibody_h3_modeling | 56 | | Score vs. rmsd | 6 | 500 | 3000 |
| | antibody_snugdock | 57 | | l_sc vs. l_rmsd | 6 | 500 | 3000 |
| Carbohydrates | glycan_dock, (dock_glycans)* | 58,59 | Jason Labonte, Morgan Nance | l_sc vs. L_rmsd | 6 | 1000 | 1100 |
| | glycan_structure_prediction | 60 | Jared Adolf-Bryfogle | Score vs. rmsd | 4 | 500 | 950 |
| Comparative modeling | RosettaCM | 61 | Jason Fell | GDT-MM | 16 | 200 | 1800 |
| Design | ddg_alanine_scan | 62 | Ajasja Ljubetič | R, MAE, fraction correctly classified | 19: 381 | 1 | 3 |
| Design | SEWING | 63 | Frank Teets | MotifScorer, InterModelMotifScorer | 1 | 100 | 75 |
| Design | enzyme_design | 64 | Rocco Moretti | Various sequence recoveries | 50 | 100 | 50 |
| Design | design_fast | 65 | Jack Maguire, Chris Bahl | Score vs. seqrec | 48 | 100 | 2600 |
| Design, interfaces | cofactor_binding_sites | 66 | Amanda Loshbaugh | rank top, position profile similarity | 7 | 200 | 170 |
| design, immune system | mhc_epitope_energy | 67 | Brahm Yachnin | Degree of de-immunization, among others | 50 | 100 | 2000 |
| docking | protein_protein_docking | 68 | Shourya SR Burman | l_sc vs. l_rmsd | 10 | 5000 | 833 |
| | ensemble docking | 69 | Ameya Hamalkar | l_sc vs l_rmsd | 3 | 5000 | 3000 |
| FlexPepDock | FlexPepDock | 70 | Ziv Ben-Aharon | reweighted l_sc vs backbone l_rmsd | 2 | 200 | 70 |
| fragments | fragment_picking | 71 | Justyna Krys, Dominik Gront | rmsd | 10 | 400 | 2000 |
| fragments | make fragments pipeline | 71 | Daniel Farrell | Coverage, precision | 65 | 1 | 3000 |
| ligand docking | ligand_docking | 50 | Shannon Smith | Delta_lsc vs. ligand_rmsd | 50 | 200 | 2000 |
| | ligand_scoring_ranking | 50 | | Spearman and Pearson correlation coefficient | 57: 285 | 1 | 2 |
| loop modeling | loop_modeling_CCD | 72 | Phuong Tran, Shane Ó Conchúir | Score vs. loop_rmsd | 7 | 500 | 500 |
| | loop_modeling_KIC | 73 | | Score vs. loop_rmsd | 7 | 500 | 620 |
| | loop_modeling_KIC_fragments | 74 | | Score vs. loop_rmsd | 7 | 500 | 760 |
| | loop_modeling_NGK | 75 | | Score vs. loop_rmsd | 7 | 500 | 570 |
| membrane protein-energy function | mp_f19_energy_landscape# | 37 | Rituparna Samanta, Rebecca Alford | ddG, depth and title angle | 4 | 1 | 10 |
| | mp_f19_decoy_discrimination | 37 | | Score vs. rmsd, Wrms | 4×100 | 1 | 2000 |
| | mp_f19_sequence_recovery | 37 | | sequence recovery, Kullback-Leibler divergence | 130 | 1 | 500 |
| | mp_f19_ddG_of_mutation | 76 | | Pearson correlation coefficient | 3 | 1 | 1 |
| membrane proteins | mp_dock | 77 | Julia Koehler Leman, Rebecca Alford | l_sc vs. l_rmsd | 10 | 1000 | 200 |
| | mp_domain_assembly | 78 | | Score vs. rmsd | 5 | 5000 | 700 |
| | mp_lipid_acc | 79 | | Accuracy | 223 | 1 | 2 |
| | mp_relax | 77 | | Score vs. rmsd | 4 | 100 | 40 |
| | mp_symdock | 77 | | l_sc vs. rmsd | 5 | 1000 | 140 |
| PDB diagnostic | PDB_diagnostic | NA | Steven Lewis, William Hansen, Sergey Lyskov | Read-in error type | entire PDB | 1 | 1000 |
| peptide structure prediction | simple_cycpep_predict | 48 | Vikram K. Mulligan | Score vs. rmsd, PNear | 1 | ~800,000 | 320 |
| | peptide_pnear_vs_ic50 | 51 | | IC50 vs. folding energy | 7 | 80,000 | 400 |
| refinement | relax_cartesian | 32 | Julia Koehler Leman | Score vs. rmsd | 12 | 100 | 120 |
| | relax_fast | 80 | | Score vs. rmsd | 12 | 100 | 120 |
| | relax_fast_5iter | 80 | | Score vs. rmsd | 12 | 100 | 120 |

**Table 1 (continued)**

| Test suite | Tests | Refs. | Test author | Quality measures | Targets | nstruct | Runtime in CPUh |
|---|---|---|---|---|---|---|---|
| RNA | rna_denovo_favorites | 81 | Andy Watkins | Score vs. rmsd | 12 | 200 | 120 |
| | stepwise_rna_favorites | 82 | | Score vs. rmsd | 12 | 200 | 240 |
| RosettaNMR | abinitio_RosettaNMR_rdc | 83 | Georg Kuenze, Julia Koehler Leman | Score vs. rmsd | 3 | 2000 | 170 |
| | abinitio_RosettaNMR_pcs | 83 | | Score vs. rmsd | 3 | 2000 | 1400 |

The number of tests is constantly being expanded. The test suite is the overall application, the test is the specific test, implemented by the test author(s). The quality measures are evaluated to choose a pass/fail criterion. The targets are the number of different proteins (or biomolecules) tested on, nstruct is the number of models built for each target, and the runtime in CPU hours is the total runtime over all targets.
*The dock_glycans test has been superceded by glycan_dock.
#The mp_f19_energy_landscape test has been renamed to mp_f19_tilt_angle.

with results and protocol captures on our website (https://graylab.jhu.edu/download/rosetta-scientific-tests/) twice per year for our users to see, download, run, and compare their own methods against. This transparency is crucial for the representation of realistic performance and to enhance the scientific reproducibility of our tools.

**Standardizing workflows highlights heterogeneity in score function implementations.** Standardizing the workflows and creating this framework provides us with the possibility of running some protocols with different score functions. Rosetta has been developed over the past 25 years and the score function has been constantly improved over this timeframe. Details of this evolution and the latest standard score function REF2015 can be found in references[35,36]. The attempt to easily switch score functions for an application reveals a major challenge: many applications employ the global default score function differently, a problem exacerbated by the various user interfaces to the code (see Supplementary Note 7). The heterogeneity in implementations makes it impossible to easily test different score functions for all of the applications and reveals that it hinders both progress and unification of the score functions, possibly into a single one.

**Use case (1): test framework allows comparison of score functions for multiple protocols.** Using our framework allows us to directly compare runs with different variables. For instance, we can compare different score functions for various applications: protein–protein docking, high-resolution refinement, loop modeling, design, ligand docking, and membrane protein ddG's (Table 2 and Figs. 3–5). We test the latest four score functions: score12, talaris2013, talaris2014, and REF2015 for all but ligand docking and membrane protein ddG's. Ligand docking has a special score function and requires adjustments—we test the ligand score function, talaris2014, REF2015, and the experimental score function betaNov2016. Membrane protein ddG's are tested on the membrane score functions mpframework2012, REF2015_mem, franklin2019, and the non-membrane score function REF2015 as a control.

The benchmark sets and quality metrics are described in Table 2 and in detail in the Supplementary Information. To compare the score functions, we plot each application's quality metrics (for instance interface score vs. interface RMSD for protein-protein docking, total score vs. loop RMSD for loop modeling). We then evaluate the "funnel quality" by computing the $P_{Near}$ metric, which falls between 0 and 1, with higher values indicating higher quality[48,49]. For the protein design test, we compute the average sequence similarity of the 10 lowest-scoring (best) models instead of $P_{Near}$ and for the membrane ddG test, we use the Pearson correlation coefficient between experimental and predicted ddG's. We further summarize the quality metrics per protocol and score function by a "winner-takes-it-all" comparison (Fig. 5A) and by an average metric overall target per application per score function (Fig. 5B).

A few main observations follow from this comparison: at first glance, in this comparison, REF2015 performs generally better overall, yet the best score function to use depends on the application—even different types of protocols can impact prediction accuracy. However, it should be noted that some tests have a small sample size due to the required computational resources, therefore impacting the statistical significance of these outcomes. Second, more recent score functions are not automatically better for any given application, likely because performance depends on how the score function was developed and tested. For a more detailed discussion, see the Supplementary Information. Third, results differ in some cases depending on

**Table 2 Tests for which we compare different score functions (score12, talaris2013, talaris2014, ref2015, ligand, betaNov16, mpframework, ref2015mem, and franklin2019), complete with quality measures, number of targets in each benchmark, number of models created (nstruct) and runtime in CPU hours per score function.**

| Test suite | Tests | score12 | ligand | mpframework | talaris13 | talaris14 | ref2015 | ref2015mem | betaNov16 | franklin2019 | Quality measures | Targets | nstruct | Runtime in CPUh |
|---|---|---|---|---|---|---|---|---|---|---|---|---|---|---|
| Docking | docking | x | | | x | x | x | | | | l_sc vs. l_rmsd | 10 | 1000 | 150 |
| Design | design_fast | x | | | x | x | x | | | | Score vs. seqrec | 48 | 100 | 2600 |
| Loop modeling | loop_modeling_CCD | x | | | x | x | x | | | | Score vs. loop_rmsd | 7 | 500 | 500 |
| | loop_modeling_KIC | x | | | x | x | x | | | | Score vs. loop_rmsd | 7 | 500 | 620 |
| | loop_modeling_KIC_fragments | x | | | x | x | x | | | | Score vs. loop_rmsd | 7 | 500 | 760 |
| | loop_modeling_NGK | x | | | x | x | x | | | | Score vs. loop_rmsd | 7 | 500 | 570 |
| Refinement | relax_fast | x | | | x | x | x | | | | Score vs. rmsd | 12 | 100 | 120 |
| | relax_fast5 | x | | | x | x | x | | | | Score vs. rmsd | 12 | 100 | 120 |
| | relax_cart | x | | | x | x | x | | | | Score vs. rmsd | 12 | 100 | 120 |
| Ligand docking | ligand_docking | | x | | | | | | x | | Delta_lsc vs. ligand_rmsd | 50 | 200 | 2000 |
| Membrane proteins | mp_ddg (ddG of mutation) | | | x | | | | | | x | Pearson correlation | 3 | 50 | 1800 |

The ligand docking and membrane ddG applications require specialized score functions.

how the data were summarized; the top-performing score functions per application from the "winner-takes-it-all" comparison are not necessarily the top performers when the average of the $P_{Near}$ value is used, as can be seen in ligand docking (Fig. 5B—reference[50] discussed this in-depth).

**Use case (2): scientific test framework facilitates bug fixes and maintenance.** The scientific test framework is also useful for code maintenance, to ensure that the correction of bugs does not invalidate the scientific performance of the application. This can be achieved by comparing the scientific performance of a run before and after fixing a bug in the code. For example, in October 2019, we identified an integer division error in one of our core libraries: the fraction 2/3 was incorrectly assumed to evaluate to 0.6666…, when in fact integer division discards remainders, yielding 0. This calculation affected the computation of hydrogen bonding energies and their derivatives and correcting it resulted in a small but perceptible change in some of the hydrogen bond energies. This led to the need to balance between fixing the bug and managing the complex interdependencies or to preserve the existing scoring behavior since the rest of the score function had been calibrated with the bug present. By running the scientific tests on a development branch in which we had fixed the bug, we confirmed that although the correction results in a small change in the energies, it had no perceptible effect on the scientific accuracy of large-scale sampling runs for structure prediction, docking, design, and any other protocol tested. This allowed us to make the correction without harming Rosetta's scientific performance. We are certain that the scientific tests will be invaluable for ensuring that future bug-fixing and refactoring efforts do not hinder the scientific performance of our software, thus illustrating a key example of scientific benchmarks informing substantive decisions developers must make as they navigate code life cycles.

**Use case (3): test framework allows detailed investigation of new score functions under development.** Our framework can also be used to test how major code improvements would affect scientific performance before they are adopted as default options in the code. As an example, we can test how newly developed score functions perform: although small molecules and proteins are generally more rigid structures, intermediate-scale molecules are frequently disordered and flexible. A recent study shows that Rosetta's estimates of rigidity (using the funnel quality metric $P_{Near}$ computed to a designed binding conformation) for peptides designed to bind to and inhibit a target of therapeutic interest correlate well with $IC_{50}$ values[51]. Since this prediction has relevance to computer-aided drug development efforts, we want to ensure that future protocol development would not impair these predictions. We created a test (called *peptide_pnear_vs_ic50*) that performs rigidity analysis on a pool of peptides that were previously characterized experimentally and computes the correlation coefficient for the $P_{Near}$ values from predicted models to the experimentally measured $IC_{50}$ values. We find that the current default score function, REF2015, produces much better predictions than the legacy talaris2013 and talaris2014 score functions ($R^2 = 0.53$, 0.53, and 0.90 for talaris2013, talaris2014, and REF2015, respectively), indicating an improvement of the score function accuracy for this particular application[35]. However, this correlation is considerably worse with the score function Beta currently under development ($R^2 = 0.19$). This reveals problems in the candidate's next-generation score function that will have to be addressed before it becomes the default. Our scientific tests embedded in the test server framework provide a means of rapidly benchmarking and addressing these problems.

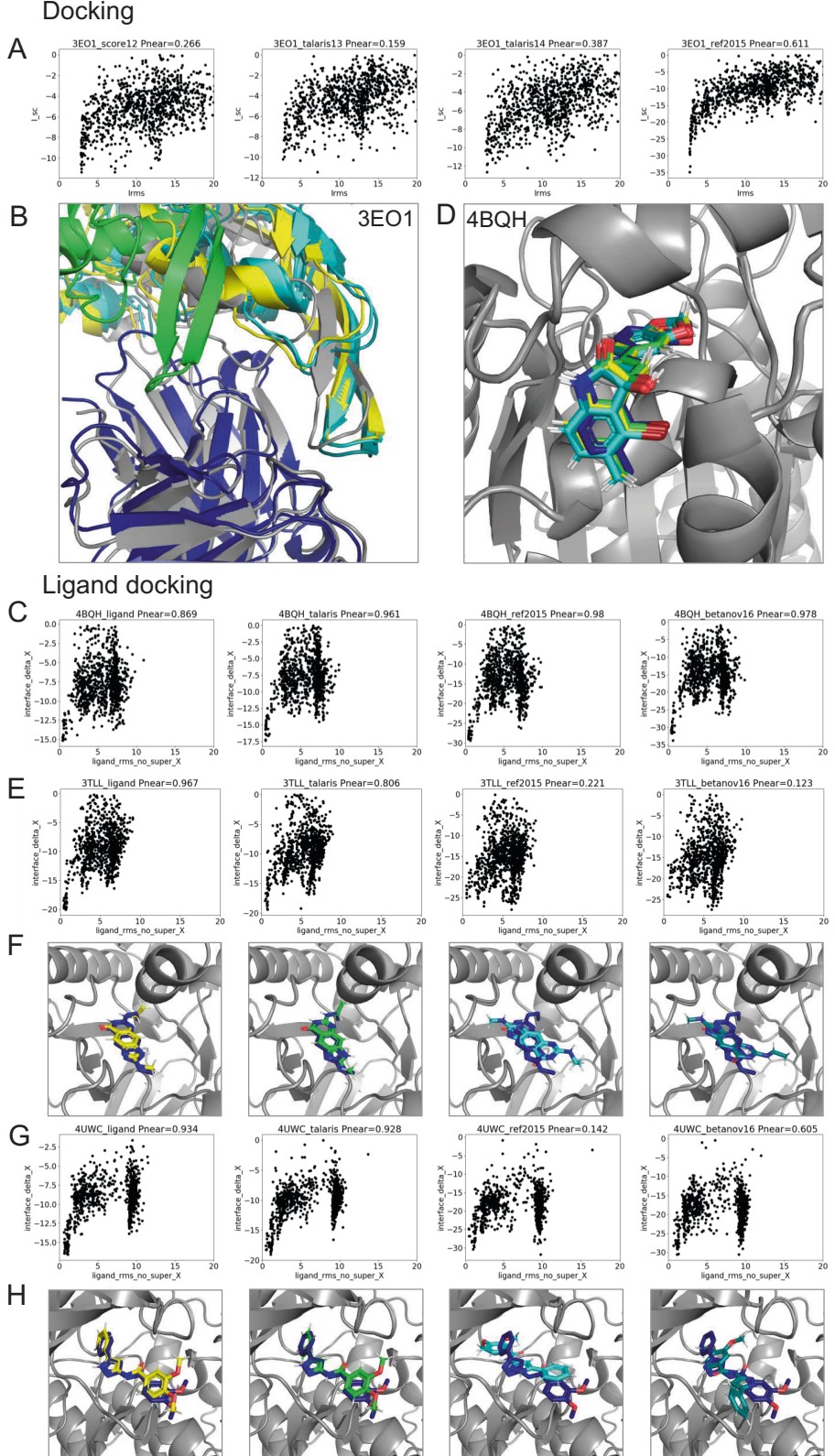

**Use case (4): this framework and tests encourage scientific reproducibility on several levels.** How is this framework useful beyond the specific tests mentioned here? Its usefulness for Rosetta developers and users lies in the protocol captures, the specific performance of each protocol, and the knowledge that scientific performance is monitored over time. Developers of macromolecular modeling methods outside of Rosetta can use

and run the exact test protocol captures to compare Rosetta's results to their own, newly developed methods. The code for the general framework to run large-scale, continuous, automated tests is available under the standard Rosetta license and is useful for developers of any type of software. Lastly, the framework highlights how software can be developed in a scientifically reproducible manner, lessons of which are useful and necessary for the

**Fig. 3 Score function comparison for specific proteins for protein–protein docking and ligand docking.** Comparison of different score functions for different applications using the $P_{Near}$ metric as an indication of "funnel quality". $P_{Near}$ falls between 0 (no funnel or incorrect global minimum) and 1 (the perfect funnel). The lambda parameter indicates the spread on the *x*-axis and is set to 4.0. Score functions are sorted from oldest to newest (left to right) and the models are colored in gray as the native (PDB) structure, then according to the score functions in order: yellow, green, cyan, and teal. **A**, **B** comparison for protein-protein docking on target with PDB ID 3eo1. The starting model is shown in dark blue—the docking partner of the starting model is too far away to be shown in the picture. The quality of the prediction improves over different score functions as indicated by tightening of the energy funnel. **C**, **D** comparison for ligand docking on target 4bqh. The native ligand pose is shown in dark blue. The quality of the prediction improves over different score functions as indicated by tightening of the energy funnel. **E**–**H** Ligand docking comparison on targets 3tll and 4uwc, respectively. Newer score functions lower the energy of an incorrect, alternative docking conformation, leading to a worse prediction.

scientific community at large. While we recognize the time and work required to implement such tests and the underlying framework, the benefits far outweigh the effort spent in trying to reproduce results that were implemented in a manner that lacks necessary aspects for reproducibility, as discussed in Table 3.

## Discussion

Here, we present a test server framework for continuously running scientific benchmarks on an integrated HPC cluster and detail the way this framework has had a positive and substantive effect on our large community of scientists. The framework itself is sufficiently general that it could in principle be used on many types of scientific software. We use it on Rosetta protocols that cover the three main interfaces to the codebase: the command line, RosettaScripts, and PyRosetta. New benchmarks are easily added and debugged, and the workflow for setting them up is well-documented and general: new tests can be added in a matter of hours and require minimal coding experience in Rosetta. We provide detailed documentation and consistency in the presentation of results, thereby facilitating maintenance by more than just experts in the community and ensuring the longevity of these tests. Automated and continuous runs of these tests allow us to recognize shifts in performance, as development is simultaneously carried out on several interdependent but otherwise unrelated fronts. Thus, we can build a longitudinal map of accuracy and scientific correctness in a constantly evolving codebase (for ourselves and our users), provide realistic protocol captures of how to run applications, and build tools that follow guidelines for improving reproducibility. Diversity in the choice of targets in the benchmark sets provides a realistic performance somewhat insulated from institutional and career incentives. So far, 40 benchmarks for various biomolecular systems and prediction tasks have been added to our server framework and more will be added over time. Due to the size of our software and the large number of protocols available, running these benchmarks requires a substantial amount of resources, which are funded through RosettaCommons, since such benchmarks are a priority for software sustainability. Even though our setup involves the integration of a custom software framework and web interface with typical HPC hardware, we expect our design choices to be of general interest and integrable with paid services such as Drone CI[44], Travis CI[45], or Jenkins[46], which are great options for small software development communities or labs that lack the hardware or personnel resources. This framework demonstrates how challenges in scientific reproducibility can be approached and handled in a general manner, even in a large and diverse community.

Implementation of a modular testing system addressing the goals above is a crucial step in achieving the reproducibility of software codes. Yet, several challenges remain that are mostly due to a lack of incentive structure. (1) In the past several years, funding agencies and journals have introduced requirements for data sharing, storing, and ensuring reproducibility. However, even if data/detailed workflows and output are shared and

available, grant or paper reviewers are likely not going to take the time to run the code because it often comes with a substantial time investment for which the reviewers do not get much in return. We argue that offering high-value incentives, such as co-authorship on the paper, mini-grants, or other compensation to the reviewer, in return for them running the code and comparing the data, could potentially make a huge difference in closing the gap in the reproducibility crisis. Alternatively, funding agencies and journals could require that another scientist, independent from the group publishing the method, is the independent code reviewer and becomes a co-author. (2) Both funding agencies and academic labs working on smaller software tools need to understand that the bulk of the work in developing a tool is not the development of the tool itself, but its maintenance, requiring years of sustained effort for it to thrive into something valuable and useful with actual impact on the scientific community. (3) Similarly, funding agencies and labs need to understand that *code is cheap but high-quality code is expensive* to create. The short-term nature of most academic research labor (undergraduate, graduate student, and postdoctoral researcher) conflicts with the long-term necessity of maintenance. Sustainable research tooling requires careful oversight and long-term management by a project leader, ensuring that maintenance responsibilities are continually reassigned as the labor pool shifts.

## Methods

The RosettaCommons community of developers has emphasized software testing for over 15 years. To support our community of hundreds of developers, our user base of tens of thousands of users, and the codebase of over 3 million lines of code[4], we implemented a custom testing architecture to fit our needs. We use this platform (a.k.a. the "Benchmark Server") to run all our tests including unit tests, integration tests, profile tests, style tests, score function tests, build tests, and others. Using this benchmark server to implement scientific tests is therefore a natural extension of its current use. Our custom testing software runs on a dedicated HPC cluster (which also runs the ROSIE server[52]), paid for by the RosettaCommons from government and non-profit funding, and commercial licensing fees.

**The backend of the benchmark infrastructure.** Our testing infrastructure consists of a number of machines:

> Database server. Our data center stores information about revisions, test, and sub-test results as well as auxiliary data like comments to revisions or a list of branches that are currently tracked via GitHub[4,53]. We are using PostgreSQL.
> Web server. The web interface for Rosetta developers connects to the database server. When a developer asks for a particular revision or test results, the webserver gathers these data from the database server, generates the HTML page, and sends it to the developer who looks at the page in a web browser. The web server also allows developers to queue new tests through the submit page on the web interface.
> Revision daemon. This application watches the state of various branches, queues tests, and sends notifications. The daemon tracks the list of branches and periodically checks if a new revision for a particular branch was committed. When a new revision has been committed, it schedules the default test set for that branch. The daemon also watches for open pull requests on GitHub, and for each pull request, it checks for specific test labels (for instance "standard tests"). The revision daemon schedules any tests with that label for that pull request.

Because scientific tests require an enormous amount of computing power, we are currently unable to test every single revision in the Rosetta main branch.

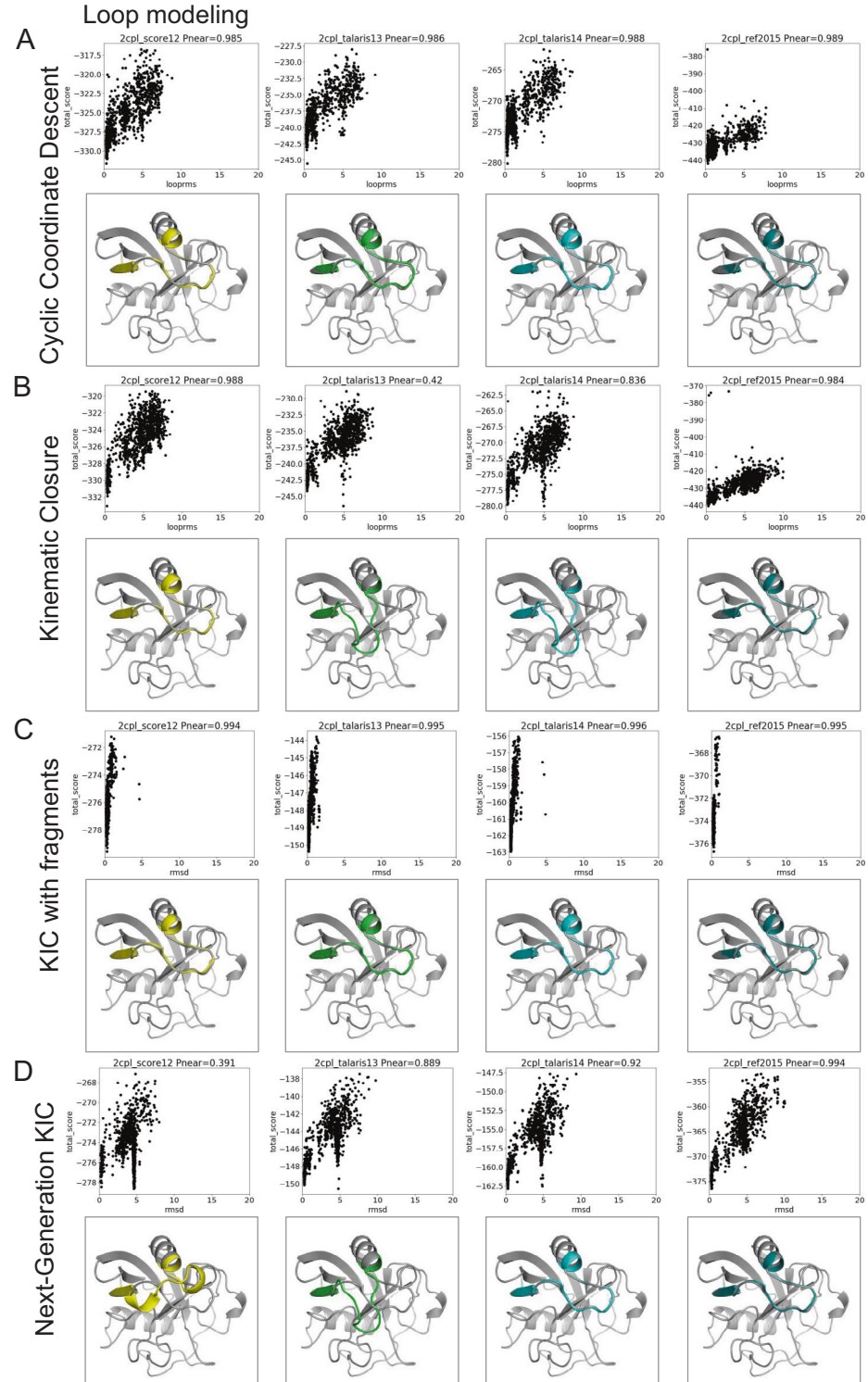

**Fig. 4 Score function comparison for one protein and different loop modeling protocols.** The protocols are **A** cyclic coordinate descent—CCD, **B** kinematic closure—KIC, **C** KIC with fragments, and **D** next-generation KIC—NGK. Score functions are sorted from oldest to newest (left to right) and the models are colored in gray as the native (PDB) structure, then according to the score functions in order: yellow—score12, green—talaris13, cyan—talaris14, and teal—ref2015. This figure shows a particularly interesting example, which is not necessarily representative of other targets. Interesting for this target are the differences in the energy landscapes that are sampled and the presence of a second, incorrect conformation with low energy for some protocols and some score functions, but not others. For 3 out of 7 targets in our comparison, including this one, most conformations that KIC (kinematic closure) with fragments samples, are close to the native structure. Again, for larger benchmarks, this is likely not as often the case.

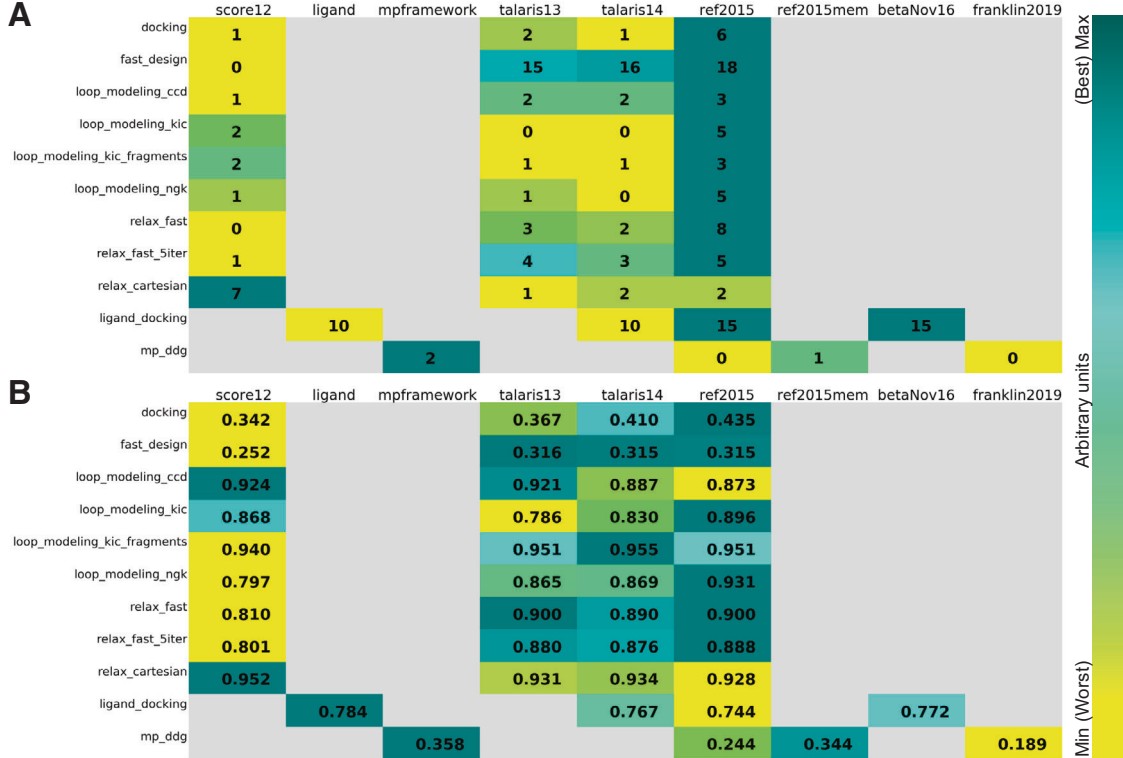

**Fig. 5 Summary of score function comparisons.** Comparison of different score functions (one per column) for different applications and protocols, using the $P_{Near}$ metric as an indication of "funnel quality". $P_{Near}$ falls between 0 (no funnel or incorrect global minimum) and 1 (the perfect funnel). The lambda parameter indicates the spread on the x-axis and is set to 4.0 in our comparison. Cells are colored according to the color bar on the right, teal is better. Unavailable data is indicated in gray. **A** The panel uses a "winner-takes-all" comparison: for each protein, the score function with the highest (i.e., best) $P_{Near}$ value (see methods) gets a point. Points are then summed by column, identifying the score function with the most and highest $P_{Near}$ values across proteins, the higher the better. **B** The averages of the $P_{Near}$ values for each score function were used, i.e., computed over each column. Higher values are better.

---

**Table 3 Guidelines for reproducible research and for the development of high-quality methods.**

| General guidelines for reproducibility | Guidelines for high-quality benchmarks |
|---|---|
| 1. Document artifacts | 1. Define scientific questions for the benchmark |
| 2. Share input, output, and exact workflow in detail under an open license in public repositories | 2. Define quality metrics that are practically relevant |
| 3. Cite the data, software, and workflows | 3. Diversify examples in the benchmark set to cover easy and difficult targets |
| 4. Use persistent links in the publication | 4. Separate benchmark set from the developed method |
| 5. Journals should check for reproducibility | 5. Pick cutting edge methods to compare your method to |
| 6. Funding agencies should fund reproducibility research | 6. Use benchmarked methods that are freely available |

---

Instead, we run scientific tests on a best-effort basis. The tests run continuously, but because there are sometimes multiple updates to the main branch per day and it takes the scientific tests about a week to run, many revisions in the main branch remain untested. In case of a test failure, the revision daemon performs a binary search bisecting the untested revisions to determine the exact revision that is responsible for the breakage.

[4.-N.] Testing daemons. The testing daemons run on various platforms: Mac, Linux, and Windows. We currently have 18 of these daemons, some of which are meant for build tests (i.e., on Windows) and some of which are capable of running tests on our HPC cluster. Each daemon periodically checks the list of queued tests from the database server. If there is any test which that daemon is capable of running, it runs the test and then uploads the test results (logs, result files, and test results encoded in JSON) to our SQL database.

This backend code is specific to our hardware, HPC use patterns, and system administration environment, and maintained separately from the code that performs or tests science. This code does not include the frontend scientific testing framework (next paragraph) and is not needed to replicate any of the scientific results. The frontend implementation of the scientific testing

framework including all the scientific benchmarks is fully available under the RosettaCommons license.

*Setup of the scientific tests.* We chose a simple setup as shown in Fig. 1C. Each scientific test requires a small number of files, available in a template directory. All files in this directory are well documented with comments, and the lines that require editing for specific tests are highlighted. Each scientific test directory starts from a template containing the following files:

- *input files*—are either located in this directory or in a parallel git submodule if the input files exceed 5 MB. This policy prevents our main code repository from becoming overly inflated with thousands of input files for scientific benchmarking.
- *0.compile.py*—compiles the Rosetta and/or PyRosetta executable.
- *1.submit.py*—submits the benchmark jobs either to the local machine *or* to the HPC cluster. Note that this "or" provides hardware non-specificity; the user writes and debugs locally and can run at scale on the benchmark server.

- *2.analyze.py*—analyzes the output data, depending on the scientific objective. Analysis functions that are specific to this particular test live in this script, while broadly useful analysis functions are located in a file that is part of the overall Python test server framework and that contains functions to evaluate quality measures.
- *3.plot.py*—plots the output data via *matplotlib*[54], or other plotting software as appropriate.
- … – other numbered scripts can be added as needed; they will run consecutively as numbered.
- *9.finalize.py*—gathers the output data and classifies the test as passed or failed, creates an HTML page by combining the documentation from the readme file, the plots of the output data and the pass/fail criterion. The HTML page is the main results page that developers, maintainers, and observers examine.
- *citation*—includes all relevant citations that describe the protocol, the benchmark set, or the quality measures.
- *cutoffs*—contains the cutoffs used for distinguishing between a pass or a failure for this test.
- *observers*—email addresses of developers that either set up the test and/or maintain it. If a test fails on the test server, an email is sent to the observers to inform them of the test breakage.
- *readme.md*—is a questionnaire-style markdown file that contains all necessary documentation to understand the purpose and detailed methods of the test. Obtaining detailed documentation is essential for the maintenance and longevity of the test. The goal is that anyone with basic Rosetta expertise and training can understand, reproduce, and maintain the test. The template readme file is provided in the Supplementary Information of this paper.

Most Rosetta protocols use the Monte-Carlo sampling protocol to create protein or biomolecule conformations, which are then evaluated by a score function.

**Reporting summary**. Further information on research design is available in the Nature Research Reporting Summary linked to this article.

## Data availability

All Rosetta code and the frontend implementation of the scientific testing framework including all the scientific benchmarks are fully available under the RosettaCommons license. In addition, complete protocol captures for all benchmarks with input files, command lines, output files, analyses, and result summaries are publicly available to view and download at https://graylab.jhu.edu/download/rosetta-scientific-tests/. These complete protocol captures are available in two code revisions and will be automatically expanded with new revisions added about every 6 months. Older revisions remain on the server. Details about each of the 42 datasets with accession codes etc. are provided under the link above.

## Code availability

Rosetta is licensed and distributed through https://www.rosettacommons.org. Licenses for academic, non-profit, and government laboratories are free of charge; there is a license fee for industry users. A license is required to gain access to the Github repository. Specific version numbers are given in the Supplementary Information.

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

## Acknowledgements

ARO MURI W911NF-16-1-0372 to Watkins; American Heart Association 18POST34080422 to Kuenze; BSF 2015207 to Schueler-Furman, Ben-Aharon; Cancer Research Institute Irvington Postdoctoral Fellowship (CRI 3442) to Roy Burman; Candian Institutes of Health Research Postdoctoral Fellowship to Yachnin; Cyrus Biotechnology to Lewis; Simons Foundation to Bonneau, Koehler Leman, Mulligan; German Research Foundation KU 3510/1-1 to Kuenze; H2020 MSCA IF CC-LEGO 792305 to Ljubetic; HHMI to Baker; Hertz Foundation Fellowship to Alford; ISF 717/2017 to Schueler-Furman, Ben-Aharon; Lundbeck Foundation Fellowship R272-2017-4528 to Stein; Mistletoe Research Foundation Fellowship to Yachnin; NCN 2018/29/B/ST6/01989 to Gront, Krys; NIAID R01AI113867 to Schief, Adolf-Bryfogle; NIEHS P42ES004699 to Siegel; NIH 1R01GM123089 to Farrell, DiMaio; NIH 2R01GM098977 to Bailey-Kellogg; NIH F31-CA243353 to Smith; NIH F31-GM123616 to Jeliazkov; NIH GM067553 to Maguire; NIH NCI R21 CA219847 and NIH R01 GM121487 to Das, Watkins; NIH NHLBI 2R01HL128537 to Yarov-Yarovoy; NIH NIAID R21 AI156570 and NIH NIBIB R21 EB028342 to Bahl; NIH NIAID U01 AI150739, NIH NIDA R01 DA046138 to Meiler, Moretti; NIH NIGMS R01 GM080403 to Meiler, Moretti and Kuenze; NIH NIGMS R01 GM073151 to Kuhlman, Gray, Leaver-Fay, Lyskov, Moretti, Meiler; NIH NIGMS R01 GM121487 and NIH NIGMS R35 GM122579 to Das; NIH NIGMS 1R01GM132110 and NIH NINDS 1R01NS103954 to Yarov-Yarovoy; NIH NINDS UG3NS114956 to Nguyen, Yarov-Yarovoy; NIH F32 CA189246 to Labonte; NIH R01 GM 076324-11 to Siegel; NIH R01 GM129261 to Woods; NIH R01 GM078221 to Harmalkar, Roy Burman, Jeliazkov, Nance, Samanta, and Gray; NIH R01 GM127578 to Gray and Labonte; NIH R01 GM110089 to Loshbaugh, Kortemme, Barlow; NIH R35 GM131923 to Leaver-Fay, Teets, Kuhlman; NIH R01 GM132565 to Hansen, Khare; NSF 1507736 to Gray, Roy Burman; NSF 1627539 and NSF 1827246 to Siegel; NSF 1805510 to Siegel, Fell; NSF 2031785 to Bahl; NSF DBI-1564692 to Loshbaugh, Kortemme, Barlow and O'Connor; NSF GRFP Fellowship to Alford; NSF CBET1923691 to Hansen, Khare; Novo Nordisk Foundation NNF18OC0033950 to Tiemann, Stein, Lindorff-Larsen; RosettaCommons Licensing Fund RC8010 to Bahl; RosettaCommons to Hansen, Moretti, Lyskov, Khare, Gray; NIH NRSA T32AI007244 and NIH U19AI117905 to Schief, Adolf-Bryfogle. The authors further thank Matt Mulqueen for expert administration of the multiple benchmark testing servers and cluster, RosettaCommons for hardware and staff support after the NIH ended their software infrastructure program, and companies that license Rosetta, providing support for critical software sustainability practices.

## Author contributions

The benchmark testing server framework was implemented and is being maintained by S. Lyskov. The scientific testing framework was created jointly by J.K.L., S. Lyskov, and S.M. Lewis. Specific tests were implemented and validated by the test authors as outlined in Table 1, namely J.J., J.W.L., M.N., J.A.B., A. Loshbaugh, F.T., R.M., J. Maguire, C.B., A. Ljubetic, B.Y., S.S.R.B., A.H., Z.B.A., J.K., D.G., D.F., S.S., P.N., J.F., S.O.C., R.S., R.A., J.K.L., S.M. Lewis, W.H., Slyskov, V.K.M., A.W., and G.K. All tests went through independent scientific and technical review by S.M. Lewis, J.K.L., S. Lyskov with help from V.K.M., R.M., A.M.W., and others for review of pull requests. Further, benchmarks were provided by J.M., C.B., K.B., S.O.C., G.K., and H.W. and independently reviewed by J.K.S.T., A.S., and K.L.L. J.J.G. supervised the creation of the benchmark infrastructure

and secured funding, together with B.K. This project was jointly supervised by R.B., J.J.G., D.G., A.L.F., C.B., C.B.K., D.B., R.D., F.D.M., S.K., T.K., J.W.L., J. Meiler, W.S., O.S.F., J.S., A.S., V.Y.Y. and B.K.

## Competing interests

Rosetta software has been licensed to numerous non-profit and for-profit organizations. Rosetta Licensing is managed by UW CoMotion, and royalty proceeds are managed by the RosettaCommons. Under institutional participation agreements between the University of Washington, acting on behalf of the RosettaCommons, their respective institutions may be entitled to a portion of the revenue received on licensing Rosetta software including programs described here. D.B., J.J.G., R.B., O.S.F., D.G., T.K., J.M., and V.Y.Y. are unpaid board members of the RosettaCommons. As members of the Scientific Advisory Board of Cyrus Biotechnology, D.B. and J.J.G. are granted stock options. S.M.L., A.L.L., and D.F. are employed by or have a relationship with Cyrus Biotechnology. Cyrus Biotechnology distributes the Rosetta software, which includes the methods discussed in this study. V.K.M. is a co-founder of and shareholder in Menten Biotechnology Labs, Inc. The content of this manuscript is relevant to work performed at Menten. J.M. is employed by Menten with granted stock options. D.B. is a cofounder of, shareholder in, or advisor to the following companies: ARZEDA, PvP Biologics, Cyrus Biotechnology, Cue Biopharma, Icosavax, Neoleukin Therapeutics, Lyell Immunotherapeutics, Sana Biotechnology, and A-Alpha Bio. CBK is a co-founder and manager of Stealth Biologics, LLC, a biotechnology company. R.B. is executive director of Prescient Design/Genentech, a member of the Roche group. The remaining authors declare no competing interests.

## Additional information

[1]Center for Computational Biology, Flatiron Institute, Simons Foundation, New York, NY 10010, USA. [2]Department of Biology, New York University, New York, NY 10003, USA. [3]Department of Chemical and Biomolecular Engineering, Johns Hopkins University, Baltimore, MD 21218, USA. [4]Cyrus Biotechnology, 1201 Second Ave, Suite 900, Seattle, WA 98101, USA. [5]Department of Immunology and Microbiology, Scripps Research, La Jolla, CA 92037, USA. [6]IAVI Neutralizing Antibody Center, Scripps Research, La Jolla, CA 92037, USA. [7]Graduate Program in Bioinformatics, University of California San Francisco, San Francisco, CA 94158, USA. [8]Department of Microbiology and Molecular Genetics, Hebrew University, Hadassah Medical School, POB 12272 Jerusalem 91120, Israel. [9]Department of Biochemistry, University of Washington, Seattle, WA 98195, USA. [10]Institute for Protein Design, University of Washington, Seattle, WA 98195, USA. [11]Genome Center, University of California, Davis, CA 95616, USA. [12]Department of Biochemistry & Molecular Medicine, University of California, Davis, CA 95616, USA. [13]Department of Chemistry, University of California, Davis, CA 95616, USA. [14]Department of Chemistry and Chemical Biology, Rutgers, The State University of New Jersey, Piscataway, NJ 08904, USA. [15]Institute for Quantitative Biomedicine, Rutgers, The State University of New Jersey, Piscataway, NJ 08904, USA. [16]Program in Molecular Biophysics, Johns Hopkins University, Baltimore, MD 21218, USA. [17]Department of Chemistry, Vanderbilt University, Nashville, TN 37235, USA. [18]Center for Structural Biology, Vanderbilt University, Nashville, TN 37235, USA. [19]Institute for Drug Discovery, Medical School, Leipzig University, 04103 Leipzig, Germany. [20]Faculty of Chemistry, Biological and Chemical Research Center, University of Warsaw, Pasteura 1, 02-093 Warsaw, Poland. [21]Department of Bioengineering and Therapeutic Sciences, University of California San Francisco, San Francisco, CA 94158, USA. [22]Biophysics Graduate Program, University of California San Francisco, San Francisco, CA 94158, USA. [23]Program in Bioinformatics and Computational Biology, University of North Carolina at Chapel Hill, Chapel Hill, NC 27599, USA. [24]Department of Physiology and Membrane Biology, School of Medicine, University of California, Davis, CA 95616, USA. [25]Chemical and Physical Biology Program, Vanderbilt University, Nashville, TN 37235, USA. [26]Department of Biochemistry and Biophysics, University of North Carolina at Chapel Hill, Chapel Hill, NC 27516, USA. [27]Linderstrøm-Lang Centre for Protein Science, Department of Biology, University of Copenhagen, DK-2200 Copenhagen N., Denmark. [28]Department of Biochemistry, Stanford University School of Medicine, Stanford, CA 94305, USA. [29]Institute for Protein Innovation, Boston, MA 02115, USA. [30]Division of Hematology/Oncology, Boston Children's Hospital, Boston, MA 02115, USA. [31]Department of Pediatrics, Harvard Medical School, Boston, MA 02115, USA. [32]Department of Computer Science, Dartmouth, Hanover, NH 03755, USA. [33]Department of Computer Science, New York University, New York, NY 10003, USA. [34]These authors contributed equally: Julia Koehler Leman, Sergey Lyskov, Steven M. Lewis. ✉email: julia.koehler.leman@gmail.com; jgray@jhu.edu; bonneau@nyu.edu

