## [Peer Review File · Nature Communications]

Ensuring scientific reproducibility in bio-macromolecular modeling via extensive, automated benchmarksReviewers' Comments:

Reviewer #1:

Remarks to the Author:

This manuscript describes how to set up a web server for the benchmarking of the Rosetta software package and its various modules. The goal of setting up such a web server is to improve the reproducibility of computer software. Generally speaking, the web server is well described and it won't be very difficult for the Rosetta software developer to follow the protocol described in this manuscript to test a newly-developed Rosetta-based module. In summary, the architecture and protocol described in this manuscript seems to be reasonable and very clear, which will definitely facilitate further development of the Rosetta software suites.

However, it is unclear if the method and protocol described in this manuscript will have a broad impact on the community or not. In order to set up a similar web server to test a software package (other than Rosetta), it will need a lot of computing resources (dedicated test server and databases) and human resources (e.g., a highly-qualified software developer). It may be affordable by a large research group but not by a small research group. That is, the method described in this manuscript may not be reproducible by small research groups.

Reviewer #2:

Remarks to the Author:

Thank you for the opportunity to read and review the manuscript "Ensuring scientific reproducibility in bio-macromolecular modeling via extensive, automated benchmarks." The paper reports the design and implementation of a test server framework, which has the goal to improve reproducibility in research based on scientific software.

I write this review from the perspective of someone who has conducted several studies on software development projects and on software platforms. I have no expertise on the scientific context of the study.

I have read the paper with great interest. I fully agree with the authors that reproducibility is a key issue in the scientific process. Given the increasing role of technology and particularly data and software in the research process, ensuring reproducibility through persistence and ongoing maintenance of underlying software systems is an important and extremely laudable goal. From the perspective of a software development researcher, the context is indeed highly interesting. The Rosetta project described in the paper shares some characteristics of an open-source project but is also specific due to the particular incentive structure for contributions and software maintenance in the scientific context. I found this fascinating, and I believe that the relevance of the problem (reproducibility in research relying on scientific software) as well as its challenging nature (e.g., complexity and specific incentive structure) make for a potentially important contribution. I also thought that the approach chosen by the project team generally makes sense and I found it interesting to read how the test server framework was designed and implemented.

Despite these strengths, I had several concerns, which mostly relate to the goal of the paper and to the presentation of the findings. Addressing these challenges seems to require substantial rewriting. I briefly discuss the concerns subsequently

First, I was struggling to fully understand what the key goal of the paper was. To me, this was the most important concern, as it speaks to the reason why the paper should be considered for publication and why readers should be interested in it. One potential goal would be to shed light on software engineering practices needed in the specific context of scientific software. This would necessarily include a discussion of what we know from other settings and what are the idiosyncrasies of the

context that warrant additional research. The authors seem to go that route when they outline the specific challenges of “running scientific benchmarks continuously.” However, it is not explicitly discussed in the remainder of the paper how these particular challenges had been addressed and on what basis it could be evaluated whether this has been done successfully. Another potential goal could be to present and describe this particular implementation of a scientific test server framework. If this was the main goal, I felt that it would be necessary to clarify why presenting the Rosetta test server framework beyond what is already documented creates value for readers.

Second, I thought that the structure of the paper was currently not fully clear, leading to somewhat loose writing and making the line of argumentation sometimes difficult to follow. Most notably, at the beginning of the results and discussion section, the authors outline a number of goals that “we think lead to more durable scientific benchmarks.” I was struggling with this formulation and the list of goals. As a reader, I would have liked to learn where these particular goals came from and why these are the most important ones. In what follows, the authors discuss these six goals but also elaborate on other aspects, such as the benchmarks that are already implemented and heterogeneity in score function implementations. Here, it would be helpful for the reader to clearly map the text to the six identified goals. In terms of structure of the paper, I was also struggling with the four use cases presented toward the end. The use cases seem important and relevant, but I was missing some arguments for why the authors present use cases at all, why these four had been selected, and what exactly we are supposed to learn from them.

Third, I thought that the authors could have done a better job in defining and explaining several terms and ideas. While some of the terminology may be clear to readers knowledgeable in the fields of bio-macromolecular modeling (which I am not) or in software engineering, I felt that defining and explaining terms could potentially make the paper much more accessible to a broader audience. This is important, especially given that the authors see their implementation as a template for scientific software in general. Increasing accessibility concerns the use cases, where the authors could have added some additional explanations. It also concerns the frontend, where the authors could have clarified what they exactly mean by some of the introduced terms.

Summing up, I think this is a valuable project and an interesting paper. I can generally see that a paper on the test server framework for the Rosetta system could provide an interesting contribution. However, it needs to be clearer what that contribution is supposed to be. Also, the writing of the paper needs to be improved, in particular in terms of the paper’s structure and the definition of key terms.

It is my hope that the authors see some value in my comments.

Thomas Kude

Reviewer #3:
Remarks to the Author:

Overall the paper is well-written and is clear and easy to follow, and describes well the current reproducibility crisis in both experimental and computational research. Over the last decade, this topic has garnered a lot of attention in computational research, with many suggestions made on how to address it.

Comments

Potentially worth the authors mentioning Sandve et al.’s ‘Ten simple rules for reproducible computational research’ (<https://doi.org/10.1371/journal.pcbi.1003285>) and discuss how/if these ideas are applicable to the benchmarks presented here.

It would also be good to comment on the growing trend of journals requiring that all code and data necessary to reproduce paper results as a prerequisite for publication, and whether simply mandating such availability is sufficient to ensure reproducibility.

The authors highlight that the methodology presented is readily applicable to other scientific software. However, the authors also acknowledge that a substantial amount of both computational resources and development time are required. It would be helpful for the authors to discuss how their approach might be applicable to smaller projects, or those with fewer or more disparate resources available.

what low-cost steps did the authors find most beneficial, and where would they recommend that developers/maintainers of an existing project start when implementing a robust testing system? Similarly, what steps would they recommend researchers take at the start of a project to ensure that this sort of robust benchmarking is easy to implement as a new project grows?

Page 11, line 329:

The effects of stochasticity on the reproducibility of computational research is an important topic. Ideally, results should be deterministic once an initial seed is specified, and the statistical properties of results should be robust with respect to random seeding. I feel the authors should expand here on why 'stochastic' failures are not considered problematic:

- Do tests fail because they do not reproducibly seed the underlying Rosetta process, or because the underlying process itself cannot be made completely deterministic through external seeding? If the former, what, if any, guidance would be appropriate for test authors when seeding their tests? If the latter, I think this is an important point to state for potential test authors to be aware of.
- Perhaps some distinction needs to be made between a test that checks for exact output (i.e. reproducible given initial seed) and statistical robustness (i.e. varying seeds leads to different output, but statistical properties are equivalent), with well-defined thresholds for what is considered 'acceptable' stochastic variation.

Page 13, line 412:

The authors comment on the effect of small sample size on the statistical significance of results. Is there any standardised approach to statistical testing (e.g. tests used, level of confidence) and, if so, was this incorporated into the benchmarks and testing guidance? If not, can you suggest appropriate guidelines on statistical testing for the scientific benchmarks?

We are grateful to the reviewers for their time investment and their excellent suggestions! We believe addressing them has elevated the quality of the paper and makes it more broadly applicable.

REVIEWER COMMENTS

Reviewer #1 (Expertise: Protein structure prediction):

This manuscript describes how to set up a web server for the benchmarking of the Rosetta software package and its various modules. The goal of setting up such a web server is to improve the reproducibility of computer software. Generally speaking, the web server is well described, and it won't be very difficult for the Rosetta software developer to follow the protocol described in this manuscript to test a newly-developed Rosetta-based module. In summary, the architecture and protocol described in this manuscript seems to be reasonable and very clear, which will definitely facilitate further development of the Rosetta software suites.

However, it is unclear if the method and protocol described in this manuscript will have a broad impact on the community or not. In order to set up a similar web server to test a software package (other than Rosetta), it will need a lot of computing resources (dedicated test server and databases) and human resources (e.g., a highly-qualified software developer). It may be affordable by a large research group but not by a small research group. That is, the method described in this manuscript may not be reproducible by small research groups.

We thank the reviewer for raising this concern. We agree that our community has its own custom setup, developed and maintained by a full-time software engineer and run on our own computing cluster. Yet, we don't think this is necessary for small groups with much less resources both computationally as well as personnel-wise. We have added a section to the beginning of the supplement and have pointed to this at the end of the introduction:

End of introduction:

“The design principles presented here can be used by anyone developing scientific software, independent of the size of the method. We highly encourage small software development groups to follow these guidelines, even though their technical and personnel setup might differ. The supplement describes several options that small groups have available to test their software with limited resources.”

Automation section:

“In small software communities that lack the ability or resources to set up a dedicated test server, integration testing via external services like Github Actions⁴³, Drone CI⁴⁴, Travis CI⁴⁵ or Jenkins⁴⁶ are an excellent alternative. More details can be found in the supplement.”

Supplement:

“Testing tools and possible setups for small software development groups

The Rosetta community is fortunate to have its own hardware and personnel resources; such resources are often not readily available for small software development groups. Below are a few recommendations of how a (similar) testing setup can be achieved with public or paid resources. The mentioned resources should serve as inspiration or starting points and we recommend checking on pricing and setup requirements, as these can change over time.

Software is often tested via Continuous Integration (CI) tests, which require software and hardware resources. Several of the automation software packages are integrated with hardware on the backend but can also be run on locally available hardware.

Automation packages:

- Github Actions¹ is integrated with Github² and a hardware backend. There is great documentation available, and one can run tests from public repositories for free. Code in private repositories can be run on a pay-per-minute basis. Setup can be achieved by a technically interested graduate student without a computer science background.
- Drone CI³ is easy to set up and is used by several well-known companies.
- Travis CI⁴ is a more mature tool. The plan is pay-per-credit and one might be able to get free compute resources for a private Github repository under an academic pro Github plan.
- Jenkins⁵ is an older tool with more complexity but might have more configuration options available. It can be run on any hardware, for instance on Amazon Web Services⁶ (AWS) through the Terraform infrastructure builder.

Hardware resources:

- Many academic labs developing software already have hardware available for compiling and running their code. Several of the automation packages mentioned above can be configured to run on this hardware.
- There are many paid cloud server options like Amazon Web Services⁶ (AWS), Heroku⁷, and Google Cloud Services⁸.
- Academic options like XSEDE⁹ funded by the US National Science Foundation or university-scale compute clusters are available to many laboratories.”

We further added a section to the end of the paper describing remaining challenges that also makes clear to funding agencies and smaller groups that the largest chunk of work in software development is not the development itself but its maintenance. We think it is important to point out because it is an often-overlooked factor, yet the reason why many software codes are underfunded and lack maintenance efforts.

“Remaining challenges to achieve scientific reproducibility

Implementation of a modular testing system addressing the goals above is a crucial step in achieving reproducibility of software codes. Yet, several challenges remain that are mostly due to a lack of incentive structure. (1) Compared to a few years ago, funding agencies and journals now have more robust requirements for data sharing, storing and ensuring reproducibility. However, even if data / detailed workflows and output are shared and available, grant or manuscript reviewers are likely not going to take the time to actually run the code because it often comes with a substantial time investment for which the reviewers don't get much in return. We argue that offering high-value incentives, such as co-authorship on the paper, mini-grants or other compensation to the reviewer, in return for them running the code and comparing the data, could potentially make a huge difference in closing the gap in the reproducibility crisis. Alternatively, funding agencies and journals could require that another scientist, independent from the group publishing the method, is the independent code reviewer and becomes a co-author. (2) Both funding agencies and academic labs working on smaller software tools need to understand that the bulk of the work in developing a tool is not the development of the tool itself, but it's maintenance, requiring years of sustained effort for it to thrive into something valuable and useful with actual impact on the scientific community. (3) Similarly, funding agencies and labs working on smaller tools also need to understand that *“code is cheap but high-quality code is expensive”* to create. Therefore, putting short-term personnel like undergraduates, postdocs or even graduate

students on a project to develop software will not make these tools sustainable once the maintenance of these tools ceases.”

Reviewer #3 (Expertise: Protein structural prediction):

Overall the paper is well-written and is clear and easy to follow, and describes well the current reproducibility crisis in both experimental and computational research. Over the last decade, this topic has garnered a lot of attention in computational research, with many suggestions made on how to address it.

Comments

Potentially worth the authors mentioning Sandve et al.'s 'Ten simple rules for reproducible computational research' (<https://doi.org/10.1371/journal.pcbi.1003285>) and discuss how/if these ideas are applicable to the benchmarks presented here.

Thanks for that suggestion. We have added this to the introduction:

“Guidelines to enhance reproducibility^{13,14} are certainly applicable and are outlined in Table 3 and are discussed in detail in an excellent editorial¹⁵ describing the *Ten Year Reproducibility Challenge*¹⁶ that is published in its own reproducibility journal *ReScience C*¹⁷.”

It would also be good to comment on the growing trend of journals requiring that all code and data necessary to reproduce paper results as a prerequisite for publication, and whether simply mandating such availability is sufficient to ensure reproducibility.

Excellent suggestion! We have added another paragraph at the end of the R&D section, including our thoughts on this.

“In the past several years, funding agencies and journals have introduced requirements for data sharing, storing, and ensuring reproducibility. However, even if data / detailed workflows and output are shared and available, grant or manuscript reviewers are likely not going to take the time to run the code because it often comes with a substantial time investment for which the reviewers don't get much in return. We argue that offering high-value incentives, such as co-authorship on the paper, mini-grants, or other compensation to the reviewer, in return for them running the code and comparing the data, could potentially make a huge difference in closing the gap in the reproducibility crisis. Alternatively, funding agencies and journals could require that another scientist, independent from the group publishing the method, is the independent code reviewer and becomes a co-author.”

The authors highlight that the methodology presented is readily applicable to other scientific software. However, the authors also acknowledge that a substantial amount of both computational resources and development time are required. It would be helpful for the authors to discuss how their approach might be applicable to smaller projects, or those with fewer or more disparate resources available.

This is a great point! It is the same point reviewer 1 made and that we have addressed. See above for our detailed response.

what low-cost steps did the authors find most beneficial, and where would they recommend that developers/maintainers of an existing project start when implementing a robust testing system?

Similarly, what steps would they recommend researchers take at the start of a project to ensure that this sort of robust benchmarking is easy to implement as a new project grows?

Developing software requires a number of elements that are necessary for success. Our goal was to communicate the six highest-priority elements to consider (shown in Fig. 1B and described in detail in the text), all of which interplay to create a system that is better than its individual part, maintainable and that allows scaling during growth of a project.

Other elements are also important and are described in detail in our Plos Comp Bio paper from last year, which has been cited multiple times throughout this manuscript (<https://journals.plos.org/ploscompbiol/article?id=10.1371/journal.pcbi.1007507>). We believe that the two papers complement each other very well in demonstrating what has worked well for our software and community and which elements to focus on.

Page 11, line 329:

The effects of stochasticity on the reproducibility of computational research is an important topic. Ideally, results should be deterministic once an initial seed is specified, and the statistical properties of results should be robust with respect to random seeding. I feel the authors should expand here on why 'stochastic' failures are not considered problematic:

- **Do tests fail because they do not reproducibly seed the underlying Rosetta process, or because the underlying process itself cannot be made completely deterministic through external seeding?** If the former, what, if any, guidance would be appropriate for test authors when seeding their tests? If the latter, I think this is an important point to state for potential test authors to be aware of.

- Perhaps some distinction needs to be made between a test that checks for exact output (i.e. reproducible given initial seed) and statistical robustness (i.e. varying seeds leads to different output, but statistical properties are equivalent), with well-defined thresholds for what is considered 'acceptable' stochastic variation.

Both. We seed the Rosetta process for integration tests but not for scientific tests because their goal is different. Integration tests are only run for a single trajectory and check that the output is the same before and after changing the code. Scientific tests check that the scientific validity of a protocol remains intact for usually thousands of trajectories without setting an initial seed. We measure scientific validity through the statistical interpretation (meaning quality metrics) of the output models. We are unable to provide specific recommendations of which statistical interpretation is the best in general because it is highly protocol-dependent. The protocol captures in the supplement provide detail on this: which quality metrics are used for specific protocols, why and how to interpret the results. We have clarified this in the maintenance section. Further, because the codebase is under constant development, changes to protocols, score functions or the underlying data structures can in rare occurrences lead to changes in trajectories, even if the random seed is set.

“Stochastic failures are an uncommon feature in software testing and are a rare but possible occurrence in this framework. Rosetta often uses Metropolis Monte Carlo algorithms and thus has an element of randomness present in most protocols. Setting specific seeds is done for integration tests in Rosetta, which are not discussed here in detail. We refrain from setting random seeds in our scientific tests because the goal is to check whether the overall statistical and scientific interpretation holds after running the same protocol twice, irrespective of the initial seed. Further, changes in the vast Rosetta codebase can cause small trajectory changes in rare cases, even with set random seeds, having a small effect on the results. Moreover, due to the reasons above, rare stochastic failures are not a concern in our case and actually point to a sensibly chosen cutoff

value. Scientific tests are interpreted in a Boolean pass/fail fashion but generally have an underlying statistical interpretation and are sampling from a distribution against a chosen target value. The statistical interpretation often varies from test to test and depends on the output of the protocol, the types of quality metrics and sample sizes; therefore, we cannot provide specific suggestions as to which statistical measures should be used in general. Details about which statistics are used in which protocol, are provided in the supplement and the linked tests. The randomness of Monte Carlo will occasionally cause a stochastic test failure because those runs happen to produce poor predictions by the tested metric. This is handled by simply rerunning the test: rare stochastic failures are not likely to occur repeatedly, and if they do, it merits re-examination of the test to change its structure or pass/fail criteria.”

Page 13, line 412:

The authors comment on the effect of small sample size on the statistical significance of results. Is there any standardised approach to statistical testing (e.g. tests used, level of confidence) and, if so, was this incorporated into the benchmarks and testing guidance? If not, can you suggest appropriate guidelines on statistical testing for the scientific benchmarks?

See above. Since the statistical interpretation is highly protocol-dependent, we are unable to generalize what should be used. The detailed supplement outlines for each of the protocols which statistical interpretation we chose and why.

Reviewer #2 (Expertise: Business informatics, collaborative software development):

Thank you for the opportunity to read and review the manuscript “Ensuring scientific reproducibility in bio-macromolecular modeling via extensive, automated benchmarks.” The paper reports the design and implementation of a test server framework, which has the goal to improve reproducibility in research based on scientific software.

I write this review from the perspective of someone who has conducted several studies on software development projects and on software platforms. I have no expertise on the scientific context of the study.

I have read the paper with great interest. I fully agree with the authors that reproducibility is a key issue in the scientific process. Given the increasing role of technology and particularly data and software in the research process, ensuring reproducibility through persistence and ongoing maintenance of underlying software systems is an important and extremely laudable goal. From the perspective of a software development researcher, the context is indeed highly interesting. The Rosetta project described in the paper shares some characteristics of an open-source project but is also specific due to the particular incentive structure for contributions and software maintenance in the scientific context. I found this fascinating, and I believe that the relevance of the problem (reproducibility in research relying on scientific software) as well as its challenging nature (e.g., complexity and specific incentive structure) make for a potentially important contribution. I also thought that the approach chosen by the project team generally makes sense and I found it interesting to read how the test server framework was designed and implemented.

Despite these strengths, I had several concerns, which mostly relate to the goal of the paper and to the presentation of the findings. Addressing these challenges seems to require substantial rewriting. I briefly discuss the concerns subsequently

First, I was struggling to fully understand what the key goal of the paper was. To me, this was

the most important concern, as it speaks to the reason why the paper should be considered for publication and why readers should be interested in it. **One potential goal would be to shed light on software engineering practices needed in the specific context of scientific software.** This would necessarily include a discussion of what we know from other settings and what are the idiosyncrasies of the context that warrant additional research. The authors seem to go that route when they outline the specific challenges of “running scientific benchmarks continuously.” However, it is not explicitly discussed in the remainder of the paper how these particular challenges had been addressed and on what basis it could be evaluated whether this has been done successfully. **Another potential goal could be to present and describe this particular implementation of a scientific test server framework.** If this was the main goal, I felt that it would be necessary to clarify why presenting the Rosetta test server framework beyond what is already documented creates value for readers.

Thank you very much for your thoughts. The goal of the paper is outlined at the end of the introduction and pertains to the implementation of the framework running the scientific tests, not the whole test server framework. The test server framework itself has not been described in a publication, so outlining it briefly at the beginning is necessary.

“... Here we address these challenges [...described above...] by introducing a general framework for continuously running scientific benchmarks for a large and increasing number of protocols in the Rosetta macromolecular modeling suite. We present the general setup of this framework, demonstrate how we solve each of the above challenges and present the results of the individual benchmarks in the supplement of this paper, complete with detailed protocol captures. The results can be used as a baseline by anyone developing macromolecular modeling methods, and the code of this framework is sufficiently general to be integrated into other types of software.”

Shedding light on software engineering practices is not the goal of this particular paper as it has been described in detail in the Plos Comp Bio paper from last year (<https://journals.plos.org/ploscompbiol/article?id=10.1371/journal.pcbi.1007507>) which we cite throughout the manuscript. We have added two sentences to the beginning of the R&D section to make this clear:

“Successful software development can be achieved by following a number of guidelines which are described in detail in reference ⁴. Software testing is an essential part of this strategy which ties into scientific reproducibility.”

Second, I thought that the structure of the paper was currently not fully clear, leading to somewhat loose writing and making the line of argumentation sometimes difficult to follow. Most notably, at the beginning of the results and discussion section, the authors outline a number of goals that “we think lead to more durable scientific benchmarks.” I was struggling with this formulation and the list of goals. As a reader, I would have liked to learn where these particular goals came from and why these are the most important ones.

We would like to respectfully refrain from rewriting large parts of this paper, as the other two reviewers and the large group of authors and find this paper easy to follow. As for where the goals are coming from, we agree, and we have added a section to the supplement describing the reasoning for these goals.

“Basis for the specific goals for the scientific test server framework

The Rosetta community had implemented scientific tests over 10 years ago which deteriorated over time. The goals outlined in the main paper are a result of critically evaluating reasons for their deterioration, combined with long-term knowledge of our community and the organization of our gigantic codebase. Below we state the goals and our reasoning behind these goals.

(1) Simplicity of the framework to encourage maintenance and support – we know from experience that more complex software design leads to higher time and labor investments in maintaining it. Simpler software design leads to more efficient support. This is critically important as we have learned from the growth over the Rosetta software suite which now has over 3.1 million lines of code.

(2) Generalization to support all user interfaces to the Rosetta codebase (command line, RosettaScripts⁴⁰, PyRosetta^{41,42}) – setting up a framework for each interface to the codebase would increase complexity and therefore maintenance and support. We wanted to support all interfaces with a single framework to make it easy on implementers and maintainers.

(3) Automation to continuously run the tests on an HPC cluster with little manual intervention – we noticed that in a huge codebase such as Rosetta, any type of automation helps tremendously in maintenance. Our community has very few people truly supporting and maintaining the software, so increasing automation and decreasing the requirement for manual intervention is necessary and useful. Additionally, a requirement for manually running these tests in a defined frequency increases the chances of them being neglected, this being the first step in deterioration.

(4) Documentation on how to add tests and scientific details of each test to allow maintenance by anyone with a general science or Rosetta background – the previous scientific tests were barely documented and therefore were very hard to maintain. The requirement to improve long-term support is proper and detailed documentation.

(5) Distribution of the tests to both the Rosetta community and their users and publicizing their existence to encourage addition of new tests and maintenance by the community. The previous scientific tests were poorly publicized, which, in addition to lack of documentation, made them almost impossible to maintain. The new tests are automatically distributed to users and developers of Rosetta and their presence on the test server website, that developers monitor frequently, improves knowledge of these tests and increases willingness to maintain them.

(6) Maintenance of the tests, facilitated by each of the previous points. Scientific tests are only useful if they're continuously run, monitored and maintained. All of the above points are a requirement to improve maintenance and support.”

In what follows, the authors discuss these six goals but also elaborate on other aspects, such as the benchmarks that are already implemented and heterogeneity in score function implementations. Here, it would be helpful for the reader to clearly map the text to the six identified goals.

In terms of structure of the paper, I was also struggling with the four use cases presented toward the end. The use cases seem important and relevant, but I was missing some arguments for why the authors present use cases at all, why these four had been selected, and what exactly we are supposed to learn from them.

Agreed - we have clarified in the subheadings which section is described, moving from addressing the 6 goals, to the results of the implementation, the use cases to describe why the implementation is useful and what it can be used for, to the remaining challenges. We strive for the busy reader to be able to grasp the content of the paper simply by reading the headings and we believe we have achieved this here. The use cases outline what this framework is useful for and what it has been used for in our community. Below are the headings in the same order as in the paper:

Goal 1 – Simplicity: Simple setup facilitates broad adoption and support from our community

Goal 2 – Generalization: New tests support interfaces of command line, PyRosetta, or RosettaScripts

Goal 3 – Automation: Tests require substantial compute power and are run on a dedicated test server

Goal 4 – Documentation: Anyone can quickly and easily add new tests

Goal 5 – Distribution: Additions and usage of tests by our community requires broad distribution

Goal 6 – Maintenance: Test failures are handled by a defined procedure

Result – Most major Rosetta protocols are now implemented as scientific benchmarks

Result – Standardizing workflows highlights heterogeneity in score function implementations

Use case #1: Test framework allows comparison of score functions for multiple protocols

Use case #2: Scientific test framework facilitates bug fixes and maintenance

Use case #3: Test framework allows detailed investigation of new score functions under development

Use case #4: This framework and tests encourage scientific reproducibility on several levels

Remaining challenges to achieve scientific reproducibility

Third, I thought that the authors could have done a better job in defining and explaining several terms and ideas. While some of the terminology may be clear to readers knowledgeable in the fields of bio-macromolecular modeling (which I am not) or in software engineering, I felt that defining and explaining terms could potentially make the paper much more accessible to a broader audience. This is important, especially given that the authors see their implementation as a template for scientific software in general. Increasing accessibility concerns the use cases, where the authors could have added some additional explanations. It also concerns the frontend, where the authors could have clarified what they exactly mean by some of the introduced terms.

This is an excellent suggestion! We have added a glossary to the end of the paper describing many Rosetta-specific terms. It would have been helpful for us to list some of the terms for us to know what is unknown but we have done our best to describe what we think might be domain specific. If any other keywords are unclear, we would like to ask the reviewer to list them and we will add them in the next round of revisions. Thank you!

“Glossary

The majority of Rosetta protocols use Monte-Carlo sampling protocol to create protein or biomolecule conformations, which are then evaluated by a score function.

PyRosetta – Python interface to the Rosetta C++ codebase; RosettaScripts – XML interface to the Rosetta C++ codebase; command line – running Rosetta executables from a terminal; observer – developers that monitor certain applications for breakage etc.; visualization – in this

context visualization of the results; RMSD – root mean square deviation: a metric to measure accuracy of predicted biomolecule models; pose – representation of the biomolecule in Rosetta; native conformation – conformation of the biomolecule as experimentally determined; GDT-MM – global distance test metric as a quality metric for prediction accuracy; MAE – mean absolute error; protocol captures – a detailed rundown of an application, complete with input, exact command lines and outputs for anyone to run applications and reproduce data; talaris2013 – score function developed in the year 2013; talaris2014 – score function developed in the year 2014.”

Summing up, I think this is a valuable project and an interesting paper. I can generally see that a paper on the test server framework for the Rosetta system could provide an interesting contribution. However, it needs to be clearer what that contribution is supposed to be. Also, the writing of the paper needs to be improved, in particular in terms of the paper’s structure and the definition of key terms.

It is my hope that the authors see some value in my comments.

Thomas Kude

Reviewers' Comments:

Reviewer #1:

Remarks to the Author:

The authors have significantly revised the manuscript, which looks better to me than the previous version.

Reviewer #3:

Remarks to the Author:

The authors have answered my concerns with regards to the paper.